# Giant single molecule chemistry events observed from a tetrachloroaurate(III) embedded *Mycobacterium smegmatis* porin A nanopore

Jiao Cao [1,2], Wendong Jia [1,2], Jinyue Zhang [1,2], Xiumei Xu[1,2,3], Shuanghong Yan [1,2], Yuqin Wang [1,2], Panke Zhang [1,2], Hong-Yuan Chen [1,2] & Shuo Huang [1,2,4]*

Biological nanopores are capable of resolving small analytes down to a monoatomic ion. In this research, tetrachloroaurate(III), a polyatomic ion, is discovered to bind to the methionine residue (M113) of a wild-type α-hemolysin by reversible Au(III)-thioether coordination. However, the cylindrical pore geometry of α-hemolysin generates shallow ionic binding events (~5–6 pA) and may have introduced other undesired interactions. Inspired by nanopore sequencing, a *Mycobacterium smegmatis* porin A (MspA) nanopore, which possesses a conical pore geometry, is mutated to bind tetrachloroaurate(III). Subsequently, further amplified blockage events (up to ~55 pA) are observed, which report the largest single ion binding event from a nanopore measurement. By taking the embedded Au(III) as an atomic bridge, the MspA nanopore is enabled to discriminate between different biothiols from single molecule readouts. These phenomena suggest that MspA is advantageous for single molecule chemistry investigations and has applications as a hybrid biological nanopore with atomic adaptors.

[1] State Key Laboratory of Analytical Chemistry for Life Sciences, Nanjing University, 210023 Nanjing, China. [2] School of Chemistry and Chemical Engineering, Nanjing University, 210023 Nanjing, China. [3] College of Physics and Electronic Engineering, Nanyang Normal University, 473061 Nanyang, China. [4] Chemistry and Biomedicine Innovation Center (ChemBIC), Nanjing University, 210023 Nanjing, China. *email: shuo.huang@nju.edu.cn

A biological nanopore, which is the core component of a commercial sequencer[1], is capable of probing the length[2], sequence[3,4], and base modifications[5] of DNA and many other biomacromolecules including RNA[6], Xeno Nucleic Acids[7], peptides[8], and proteins[9]. This remarkable sensing performance originates from its biological role as a channel protein[10]. When proper interactions are established, a biological nanopore is able to resolve a monatomic ion[11], indicating a precision far greater than that of a solid state nanopore. Pioneered since 1997 by Bayley et al.[11], nanopore-based direct sensing of single ions such as $Co^{2+}$, $Ag^+$ or $Cd^{2+}$ can be performed by designed ion-amino acid coordination[11–13] or by an ion-chelator interaction[14] within engineered α-hemolysin (α-HL) mutants. However, α-HL blockages by single monatomic ions suffer from an extremely low event amplitude (~2–3 pA), due to the cylindrical pore geometry, the small size of the ionic analyte and possible gating behaviors when monitored at a high voltage[11–13]. Alternatively, indirect sensing of metal ions can be performed with molecular adapters such as DNA[15], peptides[16] or cyclodextrins[17] but with a diminished signal specificity and an increased system complexity.

Chloroauric acid ($HAuCl_4$), a well-known Au(III) compound[18], has been widely used as a precursor for the fabrication of gold nanomaterials[19]. When dissolved in an aqueous solution, it ionizes, producing tetrachloroaurate(III) ($[AuCl_4]^-$), which is a square planar, polyatomic ion with a net charge of −1, in which the Au-Cl bond length is 2.28 Å[20]. Previous reports indicate that tetrachloroaurate(III) is a potent aquaporin inhibitor[20] but investigations at the level of a single molecule have not been reported.

With single molecule evidences, we have found that tetrachloroaurate(III) is an inhibitor of wild-type (WT) α-HL. This results from a coordination interaction established between a Au(III) atom and the thioether residue within the pore restriction. This coordination mechanism was adapted to the conical shaped *Mycobacterium smegmatis* porin A (MspA) nanopore[21,22]. A further amplified event amplitude, up to ~55 pA, was monitored.

To the best of our knowledge, single molecule study of Au(III)-thioether coordination chemistry has never been reported, and it brings insights in an aspect of bioinorganic chemistry, such as the design of Au(III) based drugs, which target proteins. The event amplitude, as generated from tetrachloroaurate(III) binding with MspA, is also the largest that has been reported from an inorganic ion when sensed by a nanopore. This suggests that MspA may be a superior template engineered as a nanoreactor to probe chemistry intermediates or kinetics in single molecule. The bound tetrachloroaurate(III) remains in the pore, forming a transient Au(III) embedment as a functional interface for sensing. As a proof of concept, the Au(III) embedded MspA nanopore discriminates between L-cysteine (Cys), L-homocysteine (Hcy) and L-glutathione (GSH) from direct single molecule readouts, which is a great challenge for fluorescence probe based imaging. It thus suggests amino acid or peptide sensing strategies with gold embedded protein nanopores or other embedments as a variety of metalloporins.

## Results

**Single tetrachloroaurate(III) binding within a WT α-HL.** The heptameric WT α-HL is a mushroom-shaped channel protein with a narrow cylindrical stem and an aperture ~1.4 nm in diameter at its narrowest spot[23]. Due to the limited acquisition bandwidth (100 kHz) of the patch clamp amplifier (Axon 200B, Molecular Devices), translocations of single inorganic ions through nanopores are not resolvable unless an interaction between the ion and the pore is established. Based on the known sulfur-gold (S-Au) coordination chemistry[24], methionine (M113)[25–28], which is in the

proximity of the 1st restriction site of the pore[29] and is the only sulfur-containing amino acid within the inner surface of an α-HL monomer, could form a reversible interaction with freely translocating tetrachloroaurate (III) ions crossing the membrane.

All electrophysiology measurements were performed with a patch clamp amplifier (Axon 200B, Molecular Devices) in an aqueous buffer consisting of 1.5 M KCl and 10 mM Tris-HCl at pH 7.0. All measurements were performed with +100 mV, continuously applied, unless otherwise stated (Methods). Chloroauric acid was added in the *cis* chamber to reach the desired final concentration. With a single WT α-HL inserted in the membrane, the anionic $[AuCl_4]^-$ is driven electrophoretically through the pore. Addition of chloroauric acid to the *cis* compartment with a final concentration of 5 μM results in a reversible current blockage, measuring 5.4 ± 0.7 pA with an average lifetime of 11 ± 1 s ($n = 3$, Supplementary Table 1, Supplementary Figs. 1 and 2). To verify this sensing mechanism, a Met → Gly mutation (M113G) was introduced by pore engineering (Supplementary Fig. 3). No $[AuCl_4]^-$ ion binding events observed with WT α-HL were detected by α-HL M113G, even when the cumulative chloroauric acid concentration in *cis* reaches 50 μM (Supplementary Fig. 4). The absence of ion binding events in the M113G mutant is thus evidence for a coordination interaction between Au(III) and the methionine (M113) in the WT α-HL.

Au(III) coordination with amino acids or peptides has been intensively investigated by UV-Vis spectroscopy, NMR spectroscopy and Fourier transform IR spectroscopy. However, these methods are limited by a lack of dynamic information, the requirement of an acidic, low-chlorine environment, a high-consumption of reactants and a lack of single molecule resolution[25–28]. With nanopores, however, the Au(III)-thioether coordination chemistry was directly monitored from single molecule readouts and with negligible requirements for the measurement environment or the quantity and the purity of the analyte. However, binding of tetrachloroaurate(III) in α-HL shows fluctuations in the trace and these lead to a wide dispersion in the statistics of the event amplitude (Supplementary Fig. 1). These fluctuations appear as low frequency transitions between different secondary states and may result from non-specific binding of tetrachloroaurate(III) with other amino acids such as lysine (K147 or K131) distributed over its long cylindrical restriction[13]. To exclude non-specific interactions, a combination of multiple mutagenesis would be necessary, and this may result in a high risk of failed pore assembly, complicating further optimizations. However, this mechanism may be adapted to other channel proteins for an enhanced sensing performance and ease of engineering.

**Methionine tetrachloroaurate(III) coordination in MspA.** In nanopore sequencing, a nanopore with a single, geometrically sharp restriction, as in MspA[21] or CsgG[30], is advantageous because it has a higher spatial resolution[3]. Though it hasn't been reported previously, direct single ion sensing could also be performed with a geometrically sharp nanopore to acquire an enlarged signal amplitude and to avoid non-specific binding with residues distant from the recognition site. The mutant M2 MspA[21] (D93N/D91N/D90N/D118R/D134R/E139K), which was the first reported nanopore for DNA sequencing, is a funnel shaped, octameric channel protein which measures ~1.2 nm in diameter at its narrowest restriction (Methods)[31]. Reported mutations, as in the M2 MspA, are designed to neutralize the original negative charges of WT MspA (PDB ID: 1uun[32]) for an enhanced capture rate for anions, such as DNA[21] or possibly tetrachloroaurate(III). Based on a visual analysis of the corresponding protein structure, no methionine or cysteine exists

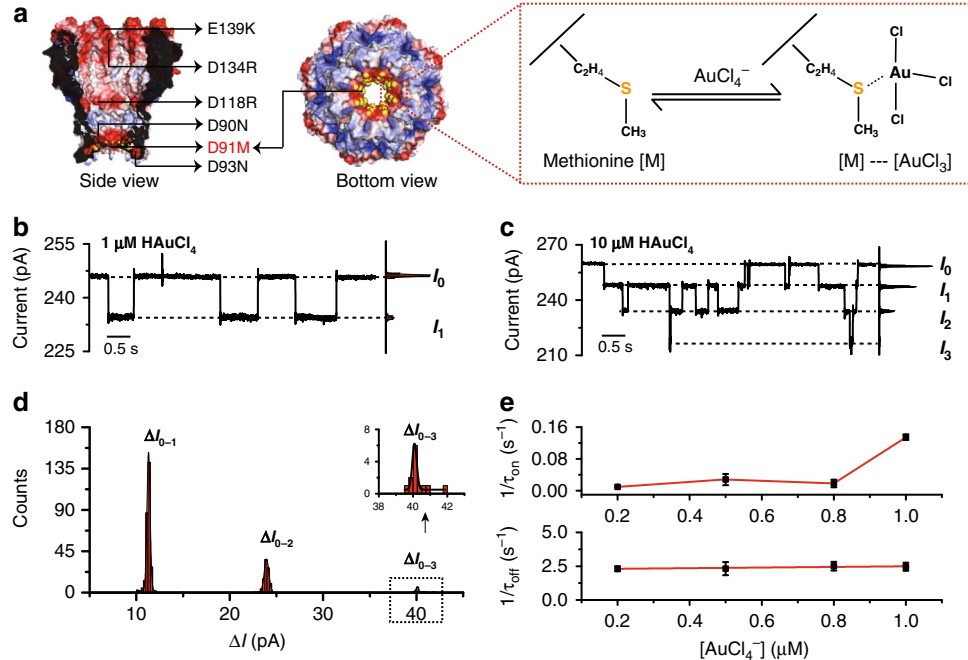

**Fig. 1 Binding of [AuCl₄]⁻ with methionine (D91M) within an engineered MspA nanopore (MspA-M). a** The structure of MspA (PDB ID: 1uun[32]) and its sensing mechanism with $[AuCl_4]^-$ ions. Pore engineering (D93N/D91M/D90N/D118R/D134R/E139K) was performed according to published methods with the exception of D91M for $[AuCl_4]^-$ sensing. The mutant MspA (MspA-M) possesses eight identical methionine residues at position 91, and capable of binding multiple $[AuCl_4]^-$ ions simultaneously. **b, c** Representative traces with all-points histogram for $[AuCl_4]^-$ sensing by MspA-M at +100 mV with 1 μM and 10 μM HAuCl₄ in *cis*, respectively. With 1 μM HAuCl₄ in *cis* (**b**), most binding events are from single $[AuCl_4]^-$ ions. Though rarely observed, it is possible to observe events resulted from simultaneous binding of two $[AuCl_4]^-$ in the same pore, which accounts for ~3% of all the events acquired. However, with 10 μM HAuCl₄ in *cis* (**c**), sequential binding events from multiple $[AuCl_4]^-$ dominate. $I_n$ stands for n $[AuCl_4]^-$ ions simultaneously in the pore. **d** Event histogram for 5 min recording for $[AuCl_4]^-$ binding by MspA-M with 10 μM HAuCl₄ in *cis*. Gaussian fitting is performed for binding events from different numbers of $[AuCl_4]^-$. $\Delta I_{0-1}$: 11.3 ± 0.2 pA, N = 280; $\Delta I_{0-2}$: 24.0 ± 0.3 pA, N = 94; $\Delta I_{0-3}$: 40.1 ± 0.2 pA, N = 13 ($\Delta I_{0-1}$ stands for the current difference between $I_0$ and $I_1$; $\Delta I_{0-2}$ stands for the current difference between $I_0$ and $I_2$; $\Delta I_{0-3}$ stands for the current difference between $I_0$ and $I_3$). **e** Plot of the reciprocals of the mean inter-event intervals ($\tau_{on}$) and plot of the reciprocals of the mean residence time ($\tau_{off}$) for single $[AuCl_4]^-$ binding events versus $[AuCl_4]^-$ ions concentration in *cis*. An abrupt increase in the event detection rates is observed at 1 μM. Further increase of HAuCl₄ concentration in *cis* leads to more sequential binding from multiple $[AuCl_4]^-$ (Supplementary Fig. 9). Mean ± Standard Deviation of $\tau_{on}$, $\tau_{off}$ are from three independent experiments (N = 3) with 10 min recording for each condition.

within the inner surface of M2 MspA, making it a clean template to which a methionine can be introduced.

Experimentally, as in α-HL M113G (Supplementary Fig. 4), no $[AuCl_4]^-$ binding events were detected by M2 MspA even when the cumulative chloroauric acid concentration in *cis* was raised to 50 μM (Supplementary Fig. 5). By a single site directed mutagenesis in M2 MspA, a methionine was introduced at residue 91 (Methods), which at 1.2 nm in diameter[22] is the narrowest spot of MspA (Fig. 1a). This MspA mutant namely MspA-M (D93N/D91M/D90N/D118R/D134R/E139K), was prepared in the same way as its predecessor (M2 MspA) (Methods) and showed similar channel properties during its characterization (Supplementary Fig. 6), indicating an octameric pore assembly, which was unaltered by the mutation.

During a continuous electrophysiology recording as described (Methods), $[AuCl_4]^-$ blockage events of MspA-M (Fig. 1b) appeared when the final concentration of HAuCl₄ in the *cis* chamber was as low as 200 nM. A significantly increased event count was observed when the concentration of HAuCl₄ in *cis* was further increased to 1 μM (Fig. 1b, Supplementary Table 2, Supplementary Video 1). Considering the existence of eight identical methionine residues as a consequence of the octameric symmetry of the pore, multi-level blockage events with approximately equal spacing gradually appeared when the HAuCl₄ concentration was increased further (Fig. 1c), indicating simultaneous blockages from more than one $[AuCl_4]^-$ within the

same pore. These blockage levels were named $I_n$, where n stands for the number of $[AuCl_4]^-$ ions in the pore simultaneously (Fig. 1b, c). Similar to $[AuCl_4]^-$ sensing with WT α-HL (Supplementary Fig. 2), only direct transitions between $I_n$ and $I_{n\pm1}$ were observed (Supplementary Fig. 7, Supplementary Video 2). This similarity between WT α-HL and MspA-M is evidence for the successful adaption of the $[AuCl_4]^-$ sensing capability from α-HL to MspA. However, even with a 40 μM concentration of HAuCl₄ in *cis*, we still observe only a maximum of three $[AuCl_4]^-$ bound to the pore restriction, possibly indicating that the pore restriction of MspA-M may be too crowded to accommodate more than 3 $[AuCl_4]^-$ simultaneously, or a repulsion between $[AuCl_4]^-$ may exist.

The event blockage amplitudes, as derived from a representative trace from a 10 min continuous recording, show fully resolved peaks, each with a Gaussian distribution (Fig. 1d) corresponding to different $I_n$ blockages. Non-specific events such as statistical counts not subject to the Gaussian distribution, are apparently never observed (Fig. 1d). Presumably, signal contributions from possible non-specific interactions distant from the restriction were weakened as negligible off-focus contributions, and this suggests a significant advantage of using a geometrically sharp channel to probe a variety of single ions or small molecules as a nanoreactor, where pore engineering is simplified to a much reduced amount of amino acids compared with that of α-HL[13]. The reduced complexity of pore functionalization by mutagenesis

of MspA thus has a great advantage in the ease of pore engineering.

Though tetrachloroaurate(III) events were detectable with a 200 nM $HAuCl_4$ concentration in the *cis*, a sharply increased detection frequency was observed (Fig. 1e) when the accumulated $HAuCl_4$ concentration in *cis* reached 1 µM. However, the $1/\tau_{off}$ value for $I_1$ events remains constant with all $HAuCl_4$ concentrations, indicating that the same type of binding was observed with the various analyte concentrations. When the $HAuCl_4$ concentration was further increased to 10 µM, $I_{n>1}$ events become dominant (Fig. 1c), from which the statistics of $\tau_{on}$ (time of inter-event duration, Supplementary Fig. 8) and the corresponding on-rate can be deduced (Supplementary Fig. 9, Supplementary Table 3). However, to avoid complications from multi-level binding events, results in Fig. 1e were limited to a low final concentration of $HAuCl_4$ in *cis*, with which 97% of $[AuCl_4]^-$ binding event results from binding of single $[AuCl_4]^-$. In a time extended measurement with up to 2 h in duration, events of $[AuCl_4]^-$ binding were stably observed, indicating that the placed $HAuCl_4$ in *cis* has rapidly reached an equilibrated state (Supplementary Fig. 10).

**Tetrachloroaurate(III) binding in different confined spaces.** Single tetrachloroaurate(III) binding has so far been demonstrated with two types of channel proteins possessing similar outer dimensions but different geometries (Fig. 2a). However, $[AuCl_4]^-$ binding within different pores may generate varying single ion behaviors when restricted differently by for example, geometry, charge or electric field in the vicinity of the restriction. These phenomena suggest how further optimization of single ion sensors may be performed, or could inspire the design of gold-containing compounds as drug molecules which target channel proteins with a high specificity and precision.

Though the experiments were performed identically, $[AuCl_4]^-$ blockage events in WT α-HL appear as shallow ($\Delta I = 5.6 \pm 0.3$ pA) resistive pulses but with a mean dwell time of ~9.38 s. $[AuCl_4]^-$ binding in MspA-M on the other hand, shows deeper blockage ($\Delta I = 11.3 \pm 0.2$ pA) but the event is shorter with a mean dwell time of ~0.45 s (Fig. 2b–d). The dwell time ($t_{off}$) follows a single exponential fitting and was derived as described in Supplementary Fig. 8. By analyzing the $\Delta I$ of $I_1$ events from WT α-HL or MspA-M, these differences were systematically observed in independent measurements by different researchers (Supplementary Tables 1 and 2). Though a larger absolute value of $\Delta I$ was observed from MspA-M, the percentages of blockage ($\Delta I/I_0$) from the two pores are quite similar. We concluded that this is an effect of the pore geometry. The narrowest regions of each pore, where tetrachloroaurate(III) binds, are similar in size (Fig. 2a) and determine the $\Delta I/I_0$. However, the wider opening of MspA, which leads to a higher open pore conductance, results in the larger $\Delta I$ in the absolute amplitude. The larger $\Delta I$ amplitude along with a considerably narrowed dispersion (Fig. 2c) shows that MspA is superior to α-HL for single ion sensing. The conical geometry results simultaneously in a faster voltage drop and a stronger electric field ($E_z = -dV/dz$). The enhanced electrophoretic force and electro-osmotic flow should contribute to the shorter duration time of tetrachloroaurate(III) binding in MspA-M (Fig. 2d).

The local charge distribution within the inner surface of the pore is critical for analyte attraction. It was found that MspA-M captures tetrachloroaurate(III) more efficiently than WT α-HL, where a 200 nM detection limit was observed with MspA-M, which is 5 times lower than that of α-HL. This may result from the positive charges introduced around the larger vestibule (D118R/D134R/E139K) of MspA-M, which was originally

designed to attract ssDNA[21]. Similar phenomena have been observed with other biological nanopores, when excessive positive charges in the pore lead to a more efficient DNA capture rate[33–35].

By measuring the voltage dependence for $I_1$ binding events, acquired with either WT α-HL (Supplementary Fig. 11a–f) or MspA-M (Fig. 2e–g, Supplementary Fig. 11g–l), a maximum $\Delta I = ~55$ pA was acquired with MspA-M when a +200 mV voltage was applied (Supplementary Fig. 11l), far beyond any pore blockage signals from a single ion that have previously been reported (Fig. 2h). For $[AuCl_4]^-$ in particular, binding at +200 mV, the MspA-M signal outweighs that from WT α-HL by ~31 pA (Fig. 2h, Supplementary Fig. 11f, l, Supplementary Table 4). On the contrary, $[AuCl_4]^-$ binding in WT α-HL generates significant baseline fluctuations, which are clearly noticeable for recordings performed with an applied voltage in excess of +100 mV (Supplementary Figs. 11 and 12) and has limited its uses at a high applied voltage for a better amplitude resolution. This phenomenon should result from undesired bindings of $[AuCl_4]^-$ with other amino acids within the cylindrical stem of α-HL. In the low voltage regime, though barely detectable ($0.90 \pm 0.08$ pA), $[AuCl_4]^-$ binding within MspA-M was still visible at an applied voltage of +20 mV (Fig. 2e). However, a minimum of +60 mV was needed for WT α-HL to resolve the events (Supplementary Fig. 11a, b). In summary, by using $[AuCl_4]^-$ as a model analyte, MspA-M clearly outperforms WT α-HL in many aspects such as a larger event amplitude, a narrower event dispersion, a higher sensing specificity and a lower detection limit. This suggests that MspA is an ideal nano-cavity with which to probe a variety of single molecule chemistry kinetics.

**Direct sensing of L-cysteine by Au(III) embedded MspA.** When bound to a methionine, the Au(III) atom remains in the proximity of the restriction of MspA for ~0.5 s, forming a transient Au(III) embedment as an adaptor for sensing. Besides the demonstrated Au(III)-thioether interaction, a stronger interaction between Au(III)-thiol is expected, as previously reported[25], which indicates that an Au(III) embedded MspA may sense a variety of thiol-containing molecules. The most abundant biothiols include L-cysteine (Cys), L-homocysteine (Hcy) and L-glutathione (GSH), which are directly involved in crucial physiological processes[36–38] such as protein synthesis[39], free radical scavenging[36] and normal immune system maintenance[40]. Though presented in the blood plasma with a high abundance, in the ~µM range[41], the structure similarity of these biothiols presents a great challenge for a direct simultaneous discrimination. With distinct physiological roles, discriminative sensing of these biothiols could have great significance in biomedical diagnostics.

Conventionally, sensing of biothiols was performed with high performance liquid chromatography-mass spectroscopy[42] or designed fluorescence probes[43], but suffers from a time consuming and laborious sample preparation process or the challenge of probe design. Nanopore sensing, which is inexpensive, fast and sensitive, may provide an alternative solution for direct sensing of biothiols. However, a biological nanopore, such as an octameric MspA-M, doesn't directly report signals for all biothiols in general (Supplementary Fig. 13) without the establishment of an interaction between the analyte and the pore.

On the other hand, the demonstrated Au(III) embedment enables MspA to interact with biothiols via the Au(III)-thiol coordination chemistry. The Au(III)-thiol coordination, which forms a much stronger bond than the established Au(III)-thioether coordination, competes with the existing Au(III)-thioether bond and consequently speeds up the dissociation of

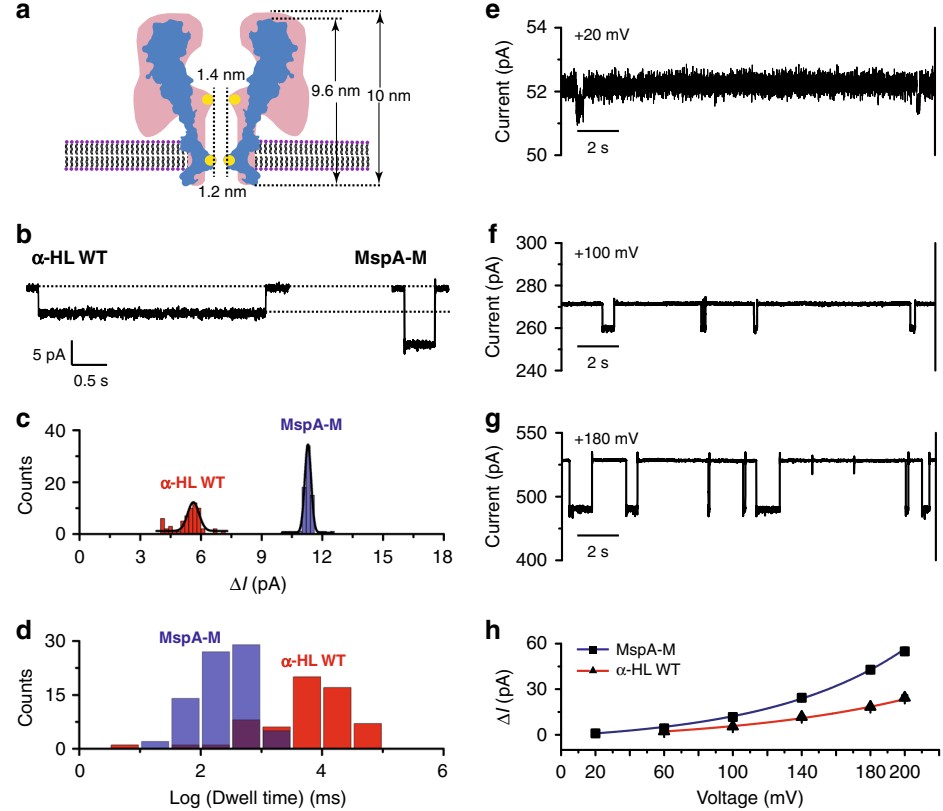

**Fig. 2 Comparison between WT α-HL and MspA-M for single [AuCl₄]⁻ sensing. a** Geometric comparison between WT α-HL (red) and MspA-M (blue). Yellow spots indicate the location of methionine in the pore. **b** Representative binding events from a single [AuCl₄]⁻ within a WT α-HL or a MspA-M nanopore at +100 mV, respectively. **c** Event-amplitude ($\Delta I$) histograms with Gaussian fitting for single [AuCl₄]⁻ bindings within a WT α-HL (red) or an MspA-M (blue) nanopore. The [AuCl₄]⁻ sensing performance is significantly improved with MspA-M showing a deeper blockage depth and a narrower distribution width (WT α-HL: 5.6 ± 0.3 pA; MspA-M: 11.3 ± 0.2 pA). **d** Log dwell-time histograms for WT α-HL (red) and MspA-M (blue) for single [AuCl₄]⁻ bindings. Single [AuCl₄]⁻ binding within MspA-M shows a systematically reduced event dwell time, possibly due to a sharper restriction from the conical MspA pore. The statistical data in **c** and **d** were taken from 10 min continuous electrophysiology recording at +100 mV with 1 μM HAuCl₄ in cis. **e–g** Representative traces for [AuCl₄]⁻ ions binding in MspA-M at +20 mV (**e**), +100 mV (**f**), and +180 mV (**g**) with 1 μM HAuCl₄ in cis. All traces (**e–g**) were digitally filtered with a 200 Hz low-pass Bessel filter (eight-pole) by Clampfit so that the shallow binding events in **e** could be presented. **h** Plot of the mean blockage depth for [AuCl₄]⁻ ions in MspA-M and WT α-HL at different voltages (Supplementary Table 4). Mean ± Standard Deviation in **h** are from three independent experiments (N = 3) with 15 min recording for WT α-HL and 5 min recording for MspA-M at each condition. An extended acquisition time for WT α-HL was taken to compensate the reduced event counts compared to that from MspA-M.

the Au(III) from the pore. Though the described chemical process happens rapidly, it can be monitored by a nanopore sensor, which forms the basis for sensing.

As a proof of concept, nanopore-based biothiol sensing was carried out with MspA-M as described in Methods. Specifically, HAuCl₄ was added to the *cis* while the biothiols were added to the *trans* compartment. The two analytes were added to different sides of a nanopore to minimize the spontaneous redox reactions between Au(III) and biothiols before entering the pore restriction (Fig. 3a). With this configuration, the anionic [AuCl₄]⁻ was first electrophoretically driven into the pore where it binds to the methionine at the pore restriction. Subsequently, the bound Au(III), which acts as an atomic bridge, captures freely translocating biothiol molecules and is stimulated to dissociate from the thioether group on the pore. Subsequently, another sensing cycle is initiated whenever the next Au(III) binds (Fig. 3b).

L-cysteine (Cys), which is an essential amino acid involved in protein synthesis[39], is the most well-known biothiol (Fig. 3c). As a test of feasibility, nanopore-based biothiol sensing was performed (Methods) by adding chloroauric acid to the *cis* and Cys to the *trans* with 4 μM and 40 μM in concentration, respectively. From electrophysiological recordings, MspA-M reported a different

type, 2-step shaped blockage event (Fig. 3d, Supplementary Video 3), which could be clearly distinguished from binding events when HAuCl₄ was the sole analyte added (Fig. 2). A representative event of such type is composed of three states namely 0, 1, and 1$_{SH}$, which corresponds to the open pore state, the [AuCl₄]⁻ bound state and the Cys bound state, as described in the molecular model from Fig. 3b. State 1, which represents the [AuCl₄]⁻ binding state (Fig. 1b), appear as a flat, less fluctuating and long residing plateau. On the other hand, state 1$_{SH}$ can be recognized from its characteristic jitter signals (Fig. 3e), which have never been observed from [AuCl₄]⁻ binding events. According to reported literatures, the interaction between the methionine residue on the pore and a [AuCl₄]⁻ is a highly reversible coordination interaction, with no redox reaction observed. This is also confirmed by the time extended measurements (Supplementary Fig. 10). However, the characteristic jitter signal might represent a redox reaction between Cys and the bound Au(III), which contains rich chemical intermediates. Nonetheless, the interaction between Au(III) and a thiol group could be indirectly monitored from the significant reduction in the dwell time of state 1 when Cys was added in *trans* (Supplementary Fig. 14). The freshly formed Au(III)-thiol interaction has stimulated the dissociation of Au(III) from the nanopore.

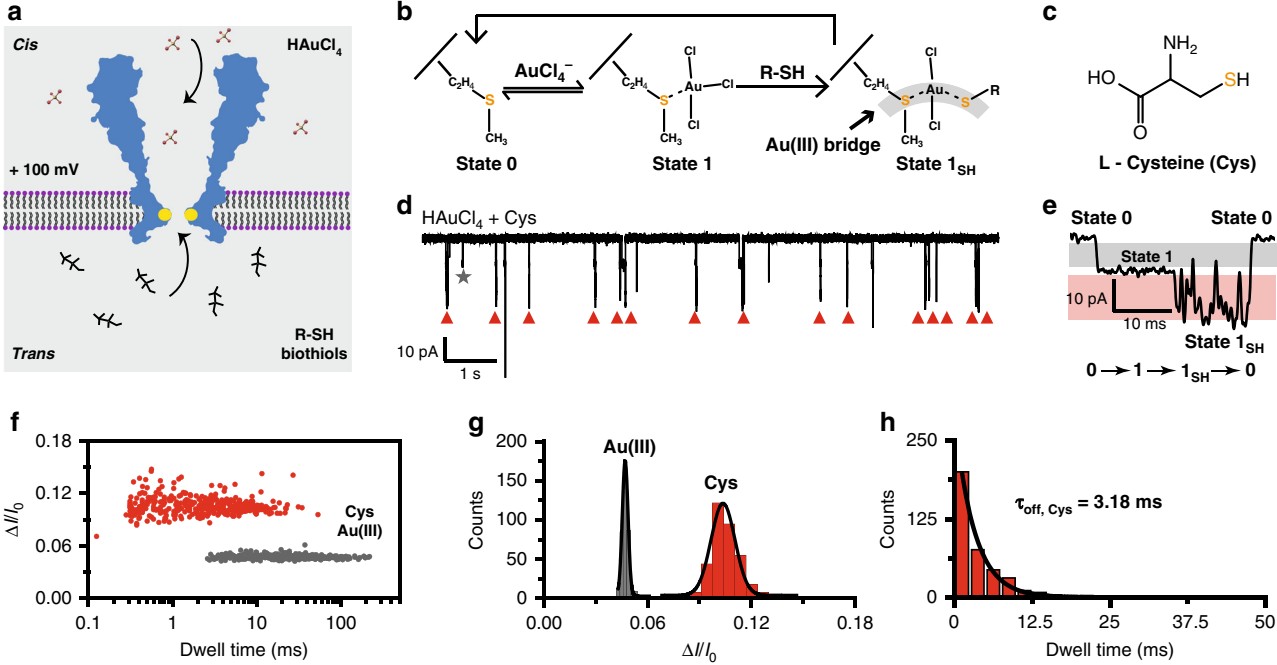

**Fig. 3 Stochastic sensing of L-cysteine by Au(III) embedded MspA-M. a** A general schematic diagram of biothiols sensing. HAuCl$_4$ were added in *cis* while the biothiols (R-SH) were added in *trans*. R represents the chemical structure of a specific type of biothiol other than the thiol group. **b** A molecular model for biothiols sensing. State 0 or 1 represents a methioine residue, without or with a [AuCl$_4$]$^-$ embedment. State 1$_{SH}$ represents a biothiol bound with the pore restriction by taking the embedded Au(III) as an atomic bridge. **c** The chemical structure of L-cysteine (Cys). L-cysteine is a specific type of biothiol. **d** Binding of Cys with [AuCl$_4$]$^-$ embedded MspA-M. The trace was recorded with a +100 mV applied voltage when 4 µM of HAuCl$_4$ in *cis* and 40 µM of Cys in *trans* were present simultaneously. Resistive pulse signals in the trace result from binding of a [AuCl$_4$]$^-$ or binding of a Cys by taking the bound Au(III) as an atomic bridge. Red triangles mark the events from Cys. The grey star marks the event of [AuCl$_4$]$^-$. Other unlabeled events are background signals as described in Supplementary Fig. 15. **e** A representative Cys blockage event. A representative Cys blockage event is composed of three states as described in **b**. State 1$_{SH}$ appears as a characteristic jitter signal. **f** The scatter plot of the relative blockade amplitude $\Delta I/I_0$ vs the dwell time from events of [AuCl$_4$]$^-$ and Cys binding. The event amplitude ($\Delta I$) for [AuCl$_4$]$^-$ or Cys is defined as the current difference between state 1 or state 1$_{SH}$ in reference to state 0 (Supplementary Fig. 16). $I_0$ is the open pore current, which was derived from the mean value of state 0. **g** The histogram of $\Delta I/I_0$ for single [AuCl$_4$]$^-$ bindings and Cys bindings. **h** The histogram of the dwell time of state 1$_{SH}$ ($t_{off,Cys}$) from a series of Cys binding events. The mean dwell time $\tau_{off,Cys}$ was derived from the single exponential fitting result. The statistical data in **f**, **g**, and **h** were from a continuous 10 min recording (Methods). (Supplementary Tables 5 and 6).

With their characteristic event shape, Cys sensing events can be immediately distinguished from other non-specific binding types. To ensure the readability, only Cys sensing events were counted in the statistics which were based on a simple algorithm that an event has to contain all three states as demonstrated in Fig. 3e. Other non-specific event types, including binding and dissociation of one [AuCl$_4$]$^-$ without any captured biothiol, sequential binding of two [AuCl$_4$]$^-$ ions and intrinsic noises from MspA-M (Supplementary Fig. 15) were ignored. Nevertheless, the Cys sensing event amounts to 96% of all detectable events from continuously recorded results (Supplementary Fig. 16). A scatter plot of the relative blockade amplitude $\Delta I/I_0$ versus the dwell time from the events of [AuCl$_4$]$^-$ and Cys sensing is shown in Fig. 3f. Though with a slightly wider dispersion than that of [AuCl$_4$]$^-$ (Fig. 3f), the Cys event forms a clear monodispersed distribution. This is clearly demonstrated in the histogram of $\Delta I/I_0$ (Fig. 3g, Supplementary Tables 5 and 6) and the dwell time $\tau_{off}$ (Fig. 3h), from which $\Delta I/I_0$ of Cys can be seen to be $0.104 \pm 0.008$ (Supplementary Table 6, $N = 364$) and the mean dwell time of state 1$_{SH}$ was derived as 3.18 ms (Fig. 3g–h).

Similar measurements were also performed with L-asparagine, L-glycine and L-glutamic acid (Supplementary Fig. 17) and no events such as that reported in Fig. 3e were observed. This indicates that this sensing configuration is highly specific to the thiol side chain of an amino acid.

**Direct sensing of L-Homocysteine by Au(III) embedded MspA.** L-Homocysteine (Hcy), which is a homologue of Cys, is an important intermediate in the metabolism of methionine and cysteine[44]. An elevated Hcy level in the blood serum indicates a high risk of cardiovascular diseases and is a critical parameter in diagnosis[45]. However, Hcy differs from Cys with just one additional methylene group (Fig. 4a), and discrimination between Hcy from Cys is a challenge.

Hcy sensing was performed as described in Fig. 3a (Methods) when 4 µM chloroauric acid was placed in *cis* and 40 µM Hcy in *trans*. Systematically deeper blockage events compared with that of Cys (~33 pA, marked by green squares) were observed from continuously recorded traces (Fig. 4b). A representative Hcy event is also composed of three states (Fig. 4c), similar to the behavior of Cys (Fig. 3e). These differ however in the 1$_{SH}$ state. A scatter plot of $\Delta I/I_0$ vs the dwell time from events of [AuCl$_4$]$^-$ and Hcy sensing was presented to show the event dispersion (Fig. 4d). From the corresponding histogram of $\Delta I/I_0$, the Hcy blockage events measure $0.13 \pm 0.01$ ($N = 330$) (Fig. 4e, Supplementary Tables 7 and 8) which confirms that a deeper relative blockage amplitude compared with that of Cys was observed. On the other hand, the mean dwell time of state 1$_{SH}$ of Hcy measures 2.94 ms (Supplementary Table 7), similar to that of Cys (Supplementary Fig. 18).

To demonstrate simultaneous discrimination between Cys and Hcy from direct single molecule readouts, a nanopore

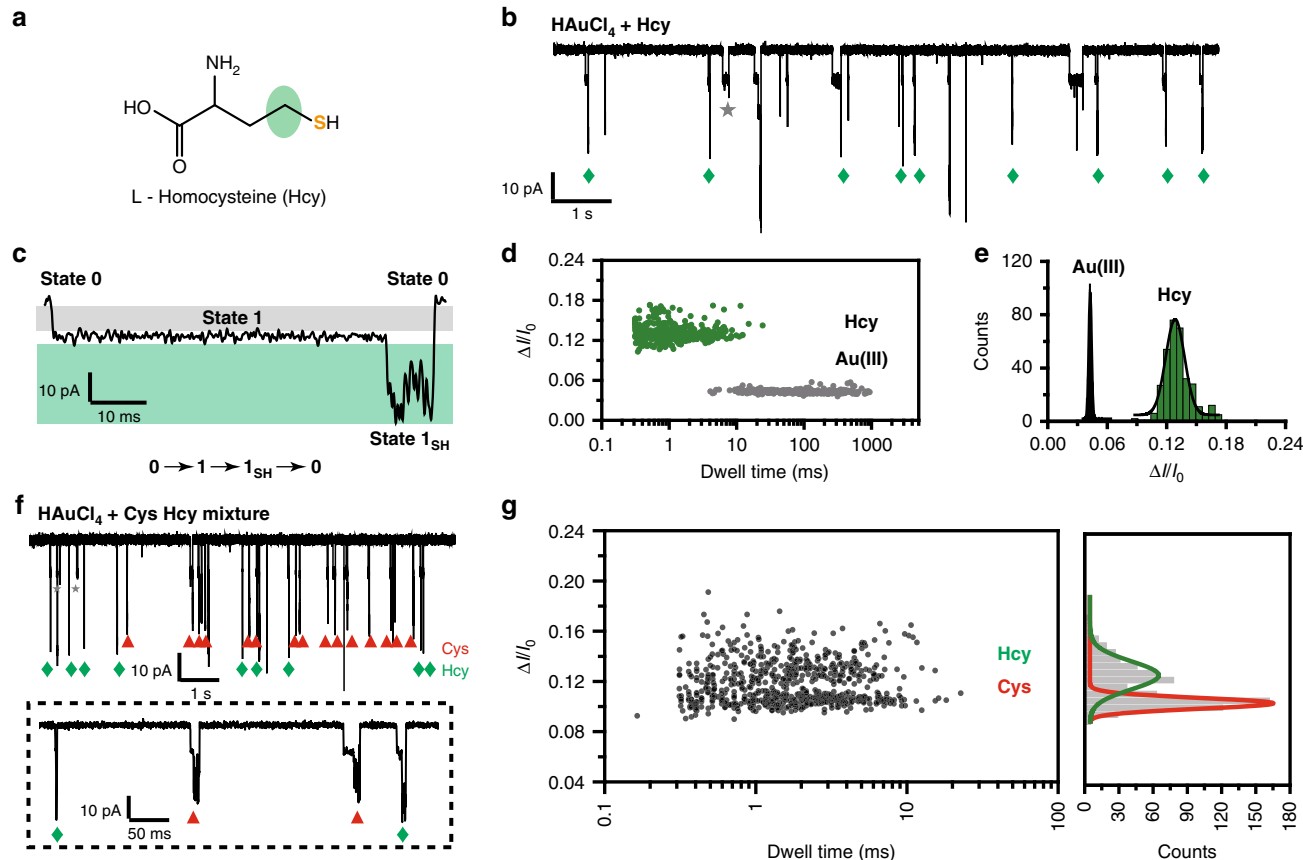

**Fig. 4 Stochastic sensing of L-homocysteine by Au(III) embedded MspA-M. a** The chemical structure of L-homocysteine (Hcy), a homologue of Cys with an extra carbon in its structure (green labeled portion). **b** Binding of Hcy with $[AuCl_4]^-$ embedded MspA-M. The demonstrated trace was recorded with a +100 mV applied voltage when 4 μM $HAuCl_4$ in *cis* and 40 μM Hcy in *trans* were present simultaneously. Resistive pulse signals in the trace result from binding of a $[AuCl_4]^-$ or binding of a Hcy by taking the bound Au(III) as an atomic bridge. Green diamonds mark the events of Hcy. The gray star marks the events of $[AuCl_4]^-$. Other unlabeled events are background signals as described in Supplementary Fig. 15. **c** A representative Hcy blockage event. Similar to Fig. 3E, a representative Hcy blockage event is composed of three states as described in Fig. 3b. Characteristic jitter signals can be recognized as state $1_{SH}$. However, the mean blockage amplitude of state $1_{SH}$ for Hcy is systematically deeper than that of Cys. **d** The scatter plot of the $\Delta I/I_0$ vs the dwell time from events of $[AuCl_4]^-$ and Hcy binding. **e** The histogram of $\Delta I/I_0$ for single $[AuCl_4]^-$ bindings and Hcy bindings. The statistical data in **d** and **e** were derived from a continuous 10 min recording with a +100 mV applied voltage (Supplementary Tables 7 and 8). **f** Simultaneous discrimination of Hcy and Cys with $[AuCl_4]^-$ embedded MspA-M. The demonstrated trace was recorded with a +100 mV applied voltage when 4 μM $HAuCl_4$ in *cis* were present. Cys and Hcy were placed in *trans* with a concentration of 20 μM for each analyte. Red triangles mark the events of Cys while green diamonds mark those of Hcy. The gray star marks the event of $[AuCl_4]^-$. A zoomed-in view of events from Hcy or Cys were demonstrated in the dotted box. **g** The scatter plot of the $\Delta I/I_0$ vs the dwell time with the corresponding amplitude histogram from Hcy and Cys sensing. The plot data were from a continuous 10 min recording as described in **f**.

measurement was performed with a mixure of 20 μM Cys and 20 μM Hcy in *trans*. The concentration of $HAuCl_4$ in *cis* was kept at 4 μM. The addition of two types of biothiols immediately reports two distinguishable event types as demonstrated from a continuously recorded trace (Fig. 4f). The scatter plot of $\Delta I/I_0$ vs the dwell time with the corresponding amplitude histogram (Fig. 4g) clearly shows two distinct event types. From the corresponding Gaussian fitting results, the two peaks in the histogram of $\Delta I/I_0$ are 0.105 ± 0.005 for Cys and 0.128 ± 0.013 for Hcy, respectively, which are in agreement with the separately measured values. The above demonstration suggests that a direct discrimination of Cys and Hcy, which differ by only one methylene group, is possible from direct nanopore readouts.

**Direct sensing of L-Glutathione by Au(III) embedded MspA.** L-Glutathione (GSH), which is a tripeptide (γ-Glu-Cys-Gly) (Fig. 5a), is critical in the maintenance of immune system. It is also the most abundant tripeptide thiol found in human serum[40].

As demonstrated with Cys and Hcy, the internal thiol in a GSH makes it compatible with the described biothiol-sensing strategy.

With 4 μM chloroauric acid in *cis* and 40 μM GSH in *trans*, characteristic biothiol-sensing events measuring ~36 pA in amplitude were observed (Fig. 5b). From a representative event, the characteristic jitter signal of GSH (state $1_{SH}$) reports a significantly larger blockage amplitude (Fig. 5c) than that of Cys (Fig. 3e). According to the negative control performed with glutamic acid and glycine separately (Supplementary Fig. 17), it was confirmed that the internal cysteine of GSH is critical to the generation of the event. The scatter plot of $\Delta I/I_0$ vs the dwell time with the corresponding amplitude histogram for GSH and $[AuCl_4]^-$ are shown in Fig. 5d. The $\Delta I/I_0$ of GSH measures 0.15 ± 0.02 (Supplementary Tables 9 and 10) with a mean dwell time of 2.82 ms in state $1_{SH}$ (Supplementary Fig. 18, Supplementary Table 9), indicating that it can be clearly distinguished from Cys via direct single molecule readouts.

Simultaneous discrimination of Cys and GSH was performed by adding a mixture of Cys and GSH in *trans* with 10 μM and

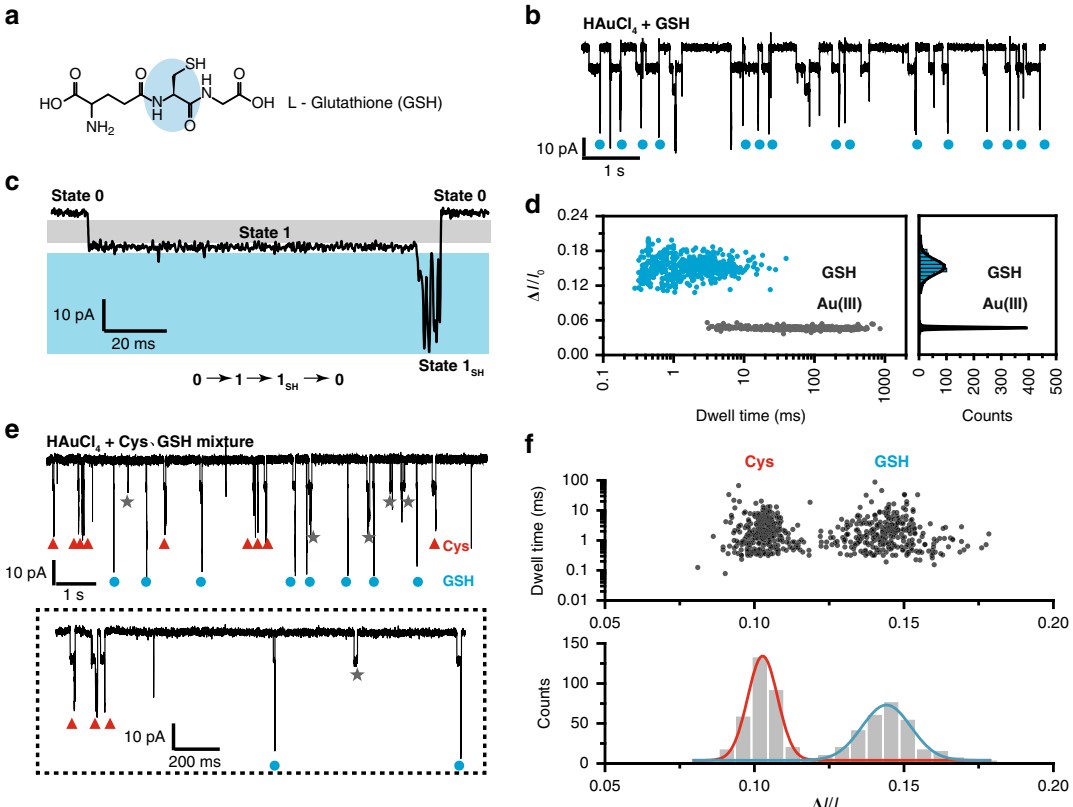

**Fig. 5 Stochastic sensing of L-Glutathione by Au(III) embedded MspA-M. a** The chemical structure of L-Glutathione (GSH). GSH is a tripeptide containing a cysteine residue (blue labeled portion). **b** Reversible and sequential binding of GSH with $[AuCl_4]^-$ embedded MspA-M. The demonstrated trace was recorded with a +100 mV applied voltage when 4 μM $HAuCl_4$ in *cis* and 40 μM GSH in *trans* were present simultaneously. Resistive pulse signals in the trace represent GSH binding with the pore. Blue dots mark the events concerning GSH. Other unlabeled events are background signals as described in Supplementary Fig. 14. **c** A representative GSH blockage event. Characteristic jitter signals similar to that of Cys or Hcy can be recognized as state $1_{SH}$. However, the mean blockage amplitude of state $1_{SH}$ for GSH is systematically deeper than that of Cys or Hcy. Gray and blue marked regions represent the the area that the transition of state 0–1 or state 1–$1_{SH}$ distributes respectively. **d** The scatter plot of $\Delta I/I_0$ vs the dwell time with the corresponding amplitude histogram from events of $[AuCl_4]^-$ or GSH binding. The definition of $\Delta I$ and $I_0$ is the same as that described in Fig. 3f. The statistical data in **d** were from a continuous 10 min recording as described in **b**. (Supplementary Tables 9 and 10). **e** Simultaneous discrimination of GSH and Cys with $[AuCl_4]^-$ embedded MspA-M. Cys and GSH were placed in *trans* with a concentration of 10 μM and 30 μM, respectively. Whereas, the concentration of $HAuCl_4$ in *cis* was kept at 4 μM. In the dotted box, a zoomed-in view of GSH and Cys events were demonstrated. Red triangles mark the events of Cys and blue dots mark those of GSH. The grey star marks binding events from one or two $[AuCl_4]^-$. **f** The scatter plot of the dwell time vs the $\Delta I/I_0$ with the corresponding amplitude histogram from the simultaneous recording of Cys and GSH as a mixture with 10 μM and 30 μM in *trans*, respectively. The statistical data were derived from a continuous 10 min recording.

30 μM final concentrations, respectively, while $HAuCl_4$ in *cis* remained at 4 μM. Compared with Cys or Hcy, GSH has a larger molecular weight and is negatively charged in a pH neutral buffer. With a +100 mV applied voltage, it was found that Cys is much more likely than GSH to be captured by the Au(III) embedded nanopore so that the GSH concentration in the mixture was increased to balance the rate of appearance of both events. From the electrophysiology trace, two types of events were clearly identified according to the difference in their amplitudes (Fig. 5e). The scatter plot of dwell time vs $\Delta I/I_0$ with the corresponding amplitude histogram for GSH and Cys are shown in Fig. 5f, from which two populations of events were clearly visible. This unambiguously confirmed that Cys and GSH can be clearly discriminated from direct nanopore readouts.

**Comparison of biothiol sensing by Au(III) embedded MspA.** The sensing events of three types of biothiols are summarized in Fig. 6a. Though the $[AuCl_4]^-$ states (state 1) appear identical, these events differ in the amplitude of their state $1_{SH}$. The

addition of biothiols in the *trans* compartment also significantly reduces the binding time of a $[AuCl_4]^-$ from a single molecule observation. Without addition of biothiols, the dwell time of a bound $[AuCl_4]^-$ measures 530 ± 90 ms. After the addition of Cys, Hcy or GSH in *trans* with a 40 μM concentration, the dwell time of state 1 was immediately reduced, which indicates that the thiol group from a biothiol molecule strongly competes with the existed Au(III)-thioether interaction. The time reduction appears to be different and this may result from the differences in steric hindrance when Cys, Hcy, or GSH interact with the Au(III). From the single molecule results, among the three tested biothiols, Cys reacts most strongly with Au(III) (Supplementary Fig. 18).

The histogram of $\Delta I/I_0$ for Cys, Hcy, and GSH events with corresponding Gaussian fittings are shown in Fig. 6b and an order of $I_{1,GSH} > I_{1,Hcy} > I_{1,Cys}$ can be clearly observed. This was expected because GSH is significantly larger than the other two amino acids and Hcy has an excess methylene when compared with Cys. From the histogram in Fig. 6b, it is clear that Cys, Hcy, and GSH can be clearly distinguished from their distinct $\Delta I/I_0$ values.

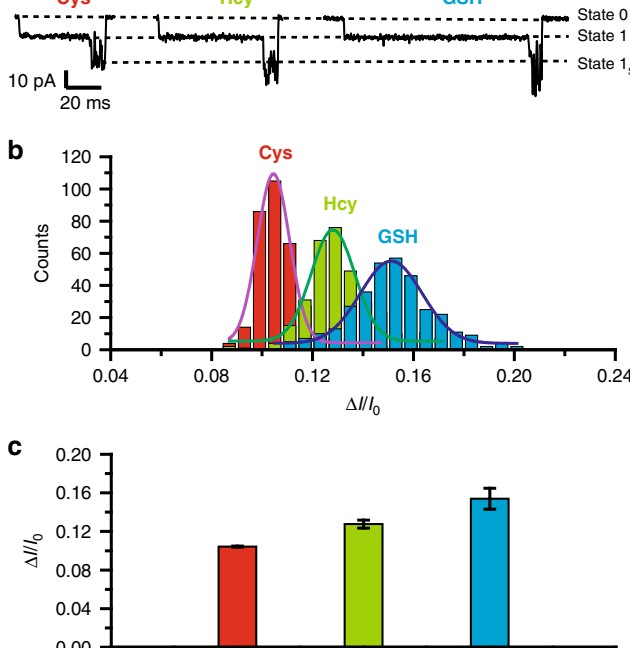

**Fig. 6 Discrimination of three types of biothiols by Au(III) embedded MspA-M. a** Representative binding events of Cys, Hcy or GSH acquired with Au(III) embedded MspA-M nanopore. **b** The histogram of $\Delta I/I_0$ with corresponding Gaussian fittings from events of Cys, Hcy, and GSH. ($N =$ 330 for each distribution). **c** The bar plot of the mean $\Delta I/I_0$ values of Cys, Hcy, and GSH. (Cys: 0.104 ± 0.0006, Hcy: 0.128 ± 0.004, GSH: 0.154 ± 0.011). All means and standard deviations were from three independent experiments for each analyte (10 min continuous recordings, Supplementary Tables 6, 8, 10).

However, due mainly to the jitter signal of state $1_{SH}$, the $\Delta I/I_0$ value appears with a wide distribution and an inevitable signal overlap. However, statistical data of mean $\Delta I/I_0$ histogram indicates that the distinction between Cys, Hcy and GSH can be achieved from independent measurements (Fig. 6c) ($n = 3$ pores for each condition, Supplementary Tables 6, 8 and 10). Besides identifying different biothiols, by calculating the proportion of events containing biothiols binding signatures with respect to all $[AuCl_4]^-$ binding events in a nanopore assay (Supplementary Fig. 19), biothiols can also be quantitatively analyzed. All above results demonstrate that the Au(III) embedded MspA could serve as a highly sensitive probe for the differentiation of GSH, Hcy and Cys.

## Discussion

Distinct from a recent report of cysteine and homocysteine discrimination using nanopores[46], in which a time consuming sample preparation is needed and GSH was in principle not detectable, the described method in this paper suggests a strategy that is simple and straightforward using nanopores with atomic adapters. Further engineering with this approach may be done by permanently embedding metal ions using irreversible coordination adaptors[47]. Full discrimination of other amino acids may be achieved with designed atomic adaptors targeting different side groups of the amino acid analytes. The locations of these adaptors within a conically shaped biological nanopore may also be slightly dispersed so that the signal amplitude from different analyte may be tuned to assist full discrimination. Similar pore engineering may be performed by taking the monomeric channel protein

OmpG[48–50] as a template for metal embedding. Other ion-amino acid combinations within a variety of biological nanopores such as Cytolysin A[51], phi29 motor protein[52] or aerolysin[2,53] could also be adapted for different applications.

To the best of our knowledge, single molecule Au(III)-methionine coordination chemistry has never been observed in a natural channel protein. It may inspire the design of gold-containing compounds as drug molecules targeting channel proteins. By single site directed mutagenesis, this mechanism has been adapted to the MspA nanopore. As a consequence of geometric optimization, MspA has magnified the event amplitude of a single $[AuCl_4]^-$ binding. The observed 55 pA $[AuCl_4]^-$ event amplitude is the largest known record from a single inorganic ion when sensed by a nanopore. The sharp restriction of MspA along with the simplicity of mutagenesis suggests its role as a superior engineering template for a variety of single molecule chemistry investigations, complementary to its well-known uses in nanopore sequencing.

With its unique physical and chemical properties, gold has been used extensively in a wide range of scientific and industrial applications such as the production of nanoparticles[54], tunneling electrodes[55], surface enhanced raman spectroscopy probes[56] and surface plasmonic resonance substrates[57]. However, these technologies lack single molecule control precision comparable to that offered by a biological nanopore, in which the geometry[58], orientation[59], polarity[51], and chemical modifications[11] of both the pore and the analyte can be manipulated and controlled. By taking the embedded Au(III) as an atomic bridge, MspA is enabled with biothiol-sensing capacities, which directly discriminate between L-cysteine, L-homocysteine, and L-glutathione from single molecule readouts. Though demonstrated as a proof of principle, this sensing mechanism is simple, label free, fast and economic and may be engineered into a portable sensor chip. Eventually, the demonstrated result of Au(III) embedment may benefit a wide range of other fundamental scientific research projects in need of a single molecule precision and the properties of gold[60,61] or even other metal elements if properly designed.

## Methods

**Materials**. Hexadecane, pentane, ethylenediaminetetraacetic acid (EDTA), Triton X-100, Genapol X-80, hydrogen tetrachloroaurate (III) hydrate (99.99%), and L-Glutathione reduced were obtained from Sigma-Aldrich. 1,2-diphytanoyl-sn-glycero-3-phosphocholine (DPhPC) was from Avanti Polar Lipids. Dioxane-free isopropyl-β-D-thiogalactopyranoside (IPTG), kanamycin sulfate, imidazole and tris (hydroxymethyl)aminomethane (Tris) were from Solarbio. 4-(2-hydroxyethyl)-1-piperazineethanesulfonic acid (HEPES) was from Shanghai Yuanye Biotechnology (China). E. coli strain BL21 (DE3) were from Biomed. LB broth and LB agar were from Hopebio (China). Hydrochloric acid (HCl) was from Sinopharm (China). L-cysteine, L-asparagine, L-glycine and L-glutamic acid were from BBI Life Sciences (China). L-Homocysteine was from J & K Chemical Technology

The potassium chloride buffer (1.5 M KCl, 10 mM Tris-HCl, pH 7.0) was prepared with Milli-Q water and membrane (0.2 μm, Whatman) filtered prior to use. Hydrogen tetrachloroaurate (III) hydrate was dissolved in Milli-Q water as a stock solution (30 mM) for subsequent experiments. L-cysteine, L-asparagine, L-glycine, L-glutamic aicd L-homocysteine, and L-glutathione reduced were dissolved in the potassium chloride buffer at 5 mM final concentration for subsequent experiments.

**α-HL preparation**. The gene coding for α-HL WT and α-HL M113G were custom synthesized and constructed in a pet 30a(+) plasmid (Genscript, New Jersey) for prokaryotic protein expression. Heptameric α-HL were expressed with E. coli BL21 (DE3) and purified with nickel affinity chromatography. After heat shock transformation with plasmid gene coding for either α-HL WT or α-HL M113G, the cells were grown in LB medium at 37 °C till $OD_{600} = 0.7$. Isopropyl β-D-thiogalactoside (IPTG) was then added to a final concentration of 1 mM for induction. After shaking overnight at 18 °C, the cells were harvested by centrifugation (2546 × g, 20 min, 4 °C). The pellet was collected and re-suspended in lysis buffer 1 (0.5 M NaCl, 20 mM HEPES, 1% Triton X-100, pH = 8.0), sonicated for 15 min and then centrifuged (18,800 × g, 4 °C, 40 in) to remove intact cells. After syringe filtration, the supernatant was loaded onto a nickel affinity column (HisTrapTM HP, GE Healthcare). After washing the column with buffer A1 (0.5 M NaCl, 20 mM

HEPES, 5 mM imidazole pH 8.0), the α-HL heptamers were then eluted with a linear gradient of imidazole to buffer B1 (0.5 M NaCl, 20 mM HEPES, 500 mM imidazole, pH 8.0). Fractions of interests were further characterized and confirmed with SDS-polyacrylamide gel electrophoresis (Supplementary Fig. 3). Only fractions of the α-HL heptamer were collected for subsequent electrophysiology.

**MspA preparation**. The gene coding for M2 MspA (D93N/D91N/D90N/D118R/ D134R/E139K) and MspA-M (D93N/D91M/D90N/D118R/D134R/E139K) were custom synthesized and constructed in a pet 30a(+) plasmid (Genscript, New Jersey) for prokaryotic protein expression. After heat shock transformation with plasmid gene coding for either M2 MspA or MspA-M, the cells were grown in LB medium to an $OD_{600} = 0.7$, induced by 1 mM isopropyl β-D-thiogalactoside (IPTG) and shaken overnight at 16 °C. The cells were harvested by centrifugation (2546 × g, 20 min, 4 °C) and the pellet was re-suspended in lysis buffer 2 (100 mM $Na_2HPO_4/NaH_2PO_4$, 0.1 mM EDTA, 150 mM NaCl, 0.5% (w/v) Genapol X-80, pH 6.5), and heated to 60 °C for 10 min. The suspension was cooled on ice for 10 min and centrifuged at 4 °C for 40 min at 16,200 × g. After syringe filtration, the supernatant was applied to a nickel affinity column (HisTrapTM HP, GE Healthcare). After washing the column with buffer A2 (0.5 M NaCl, 20 mM HEPES, 5 mM imidazole, 0.5% (w/v) Genapol X-80, pH = 8.0), bound proteins were eluted with a linear gradient of imidazole to buffer B2 (500 mM imidazole, 0.5 M NaCl, 20 mM HEPES, 0.5% (w/v) Genapol X-80, pH = 8.0). The fractions for MspA octamer were collected and characterized by 12% SDS-PAGE (Supplementary Fig. 6) and used directly for electrophysiology measurements.

**Electrophysiology recording and data analysis**. All electrophysiology results were acquired by an Axopatch 200B patch clamp amplifier and digitized by a Digidata 1550 A1 digitizer (Molecular Devices, UK). A custom made measurement chamber is separated by a Teflon film (30 μm thick) with an orifice (ø = 100 μm). Before use, the orifice was pretreated with 0.5% (v/v) hexadecane in pentane and then air-dried to evaporate the pentane. 1,2-Diphytanoyl-sn-glycero-3-phos-phocholine (DphPC) was used to form a self-assembled lipid bilayer sealing the orifice. This lipid bilayer divides the chamber into *cis* and *trans* compartments both filled with 0.5 mL of 1.5 M KCl buffer (1.5 M KCl, 10 mM Tris-HCl, pH 7.0). A pair of Ag/AgCl electrodes were placed in *cis* and *trans* side of the chamber, in contact with the aqueous buffer respectively. Conventionally, the *cis* side is defined as the side which is electrically grounded. Biological nanopores (WT α-HL, α-HL M113G, M2 MspA, or MspA-M) were added to *cis* for spontaneous pore insertion.

Unless otherwise stated (Supplementary Figs. 11 and 12), all measurements were performed with a +100 mV continuously applied voltage. The acquired single channel data was sampled at 25 kHz and filtered with a corner frequency of 1 kHz. For $[AuCl_4]^-$ binding events, the recorded current traces were digitally filtered with a 200 Hz low-pass Bessel filter (eight-pole). For sensing of amino acids, the recorded current traces were digitally filtered with a 0.2 kHz low-pass Bessel filter (eight-pole). Event states were detected by the single channel search feature in Clampfit 10.7 and further analyses (histogram, curve fitting and plotting) were carried out in Origin 9.1 (Origin Lab).

## Data availability
The data that supports the results within this paper are available from the corresponding authors upon reasonable request.

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

## Acknowledgements

The authors would like to thank Academician/Prof. Zijian Guo (Nanjing University), Prof. Jin Zhao (Nanjing University) and Prof. Yuncong Chen (Nanjing University) for inspiring discussions on the coordination chemistry of Au(III) complexes. Prof. Dennis Gillingham (University of Basel) for the suggestion of the manuscript. Prof. Hagan Bayley (University of Oxford), Prof. Shaolin Zhu (Nanjing University) and Prof. Congqing Zhu (Nanjing University) for inspiring discussions. This work was funded by National Natural Science Foundation of China (Grant No. 91753108, No. 21327902, No. 21675083, No. 31972917), Fundamental Research Funds for the Central Universities (Grant No. 020514380142, No. 020514380174), State Key Laboratory of Analytical Chemistry for Life Science (Grant No. 5431ZZXM1804, No. 5431ZZXM1902), Excellent Research Program of Nanjing University (Grant No. ZYJH004), 1000 Plan Youth Talent Program of China, Programs for high-level entrepreneurial and innovative talents introduction of Jiangsu Province. Technology innovation fund program of Nanjing University.

## Author contributions

S.H. conceived the project and designed the experiments. J.C., W.D.J., J.Y.Z., S.H.Y., and X.M.X. performed the experiments. J.C., W.D.J., and S.H.Y. expressed and purified protein nanopores. P.K.Z. and Y.Q.W. set up the instruments. S.H. and J.C. analyzed the data and wrote the paper. S.H. and H.Y.C. supervised the project.

## Competing interests

S.H. and J.C. are inventors on a filed PCT patent application related to this work (PCT/CN2019/102756, Aug/28/2018). The authors declare no other competing interest.
