## [Peer Review File · Nature Communications]

Reviewers' Comments:

Reviewer #1:

Remarks to the Author:

Nanochannels either of biological origin in soft membranes or drilled in solid material are used as spacer between two aqueous compartments. Analysing the time dependent ion current across nanopores allows to identify small molecules or even to detect the composition of long DNA molecules. For this the constriction in the nanopore is the critical element and subsequently one crucial part in the design is to modify the orifice to be suited to create a good signal to noise needed for detection.

Here the authors suggest a clever combination: tetrachloroaurate(III) binds strongly to methionine. This is a smart way to reduce the pore size and the presence of gold creates additionally a strong affinity to thiol groups. The authors show the effect by genetic engineering in an otherwise methionine free protein channel. Although simple in concept it hasn't yet been applied.

The authors compare the efficiency of the concept for two typical channels: MspA and alpha-hemolysin. MspA is octo-oligomer and mutagenesis will give a single zone of methionine. The authors test their system with three different thiol-containing biomolecules.

The work is carefully done and interesting to a wider community.

Below are a few points the author might wish to consider:

- Fig 1b shows binding events at low Au-salt: there should also be (a few, but some) multiple steps? The rhs shows a tenfold Au concentration with more (and multiple) steps. What happens at higher concentrations? Should there be a saturation? With 8 binding sites in very close vicinity I expect some co-operativity. How Fig 1e will look like at higher Au concentration? Could the binding site be a nucleation site for precipitation and Au particle growth?

Would pre-incubation lead to a more equilibrated system?

- Could the recordings with thiols be quantitative?

- Fig.2. the figures should be visible also in black and white, please use different symbols

Reviewer #2:

Remarks to the Author:

This research successfully utilizes the alpha-hemolysin aHL and MspA protein channels to clearly demonstrate single molecule Au(III)-thioester and Au(III)-thiol coordination chemical reactions and combine these reactions to discriminate single molecule biothiol compounds L-cysteine, L-homocysteine and L-glutathione from. The first finding is that the anion tetrachloroaurate (AuCl_4^-) can selectively bind to the thioester of methionine constructed in the lumen of both protein pores. The reversible binding/release of a single AuCl_4^- to/from one methionine can be clearly read out from the nanopore conductance variation. The second finding is that MspA is superior compared with the aHL pore in terms of AuCl_4^- binding efficiency, event conductance dispersion, noise levels and blocking levels, due to the native conical shape lumen of the MspA pore that features a sharp sensing site at the trans opening. The third and the most important finding is that the Au(III) on the methionine can be further attacked by various biothiol compounds. When binding, the Au-thiol coordination with biothiol compounds is stronger than the Au-thioester bond with methionine, thus quickly removing the Au-thiol compound from the methionine. In this reaction, the intermediate structure for methionine-Au(III)-thiol compounds can block the nanopore characteristically, thus allowing discrimination of three biothiol compounds at the single molecule levels. This work is significant and has impact in at least two aspects: it demonstrates the capability of nanopore to dissect the mechanism for single molecule metal ion (e.g. gold)-involved chemical reaction with biological substances; and through manipulation of different Au-thioester and Au-thiol reactions to detect different biothiol compounds and potentially proteins. The paper overall is suitable for publishing in this journal, but the authors still need to address several questions as follow

1. The authors firstly used the Au-thioester reaction to capture a tetrachloroaurate (III) (Au(III)) onto methionine on the nanopore, then used the Au-thiol reaction to capture biothiol compounds. The Au-thiol reaction can replace Au-thioester reaction, removing Au from methionine. To support this mechanistic hypothesis, it would be necessary to test the thioester compounds such as methionine. This free methionine in the solution should compete with the methionine on the nanopore to capture Au (III) on the binding site. This reaction should be quite different from the Au-thiol reaction.

2. Au(III) binding to methionine on the nanopore surface is a one-step reaction. This can be reflected by the one-step nanopore current blockade. In contrast, the binding of biothiol compound to the bound Au(III) is a multi-step binding, as reflected by the dramatic multi-level current variation at the end of each event. These are intermediate states. Can the authors discuss this difference between the two types of Au-S reactions and their nanopore signatures.

Response Letter

Reviewer #1 (Remarks to the Author):

Review Comments:

Nanochannels either of biological origin in soft membranes or drilled in solid material are used as spacer between two aqueous compartments. Analysing the time dependent ion current across nanopores allows to identify small molecules or even to detect the composition of long DNA molecules. For this the constriction in the nanopore is the critical element and subsequently one crucial part in the design is to modify the orifice to be suited to create a good signal to noise needed for detection.

Here the authors suggest a clever combination: tetrachloroaurate(III) binds strongly to methionine. This is a smart way to reduce the pore size and the presence of gold creates additionally a strong affinity to thiol groups. The authors show the effect by genetic engineering in an otherwise methionine free protein channel. Although simple in concept it hasn't yet been applied.

The authors compare the efficiency of the concept for two typical channel: MspA and alpha-hemolysine. MspA is octo-oligomer and mutagenesis will give a single zone of methionine. The authors test their system with three different thiol-containing biomolecules. The work is carefully done and interesting to a wider community. Below a few points the author might wish to consider:

Author Feedbacks:

We highly acknowledge the reviewer for such positive feedbacks regarding our manuscript. Engineered biological nanopores are able to resolve binding of single molecules or reactive intermediates in the pore lumen, showing an exceptional sensing resolution. Studies on this topic may inspire new discoveries in bioinorganic chemistry and biochemistry or may suggest new design strategies for drug molecules.

As also commented by the reviewer, the core finding of this manuscript is the pore construction and the single molecule chemistry that was monitored in its nano-cavity. Besides, the demonstrated experiment is rather simple to be carried out. For such reason, we decided to share the plasmid DNA of MspA-M in a public repository for other colleagues interested in relevant studies.

We also acknowledge the reviewer for other comments and questions, which are extremely visionary and constructive. We have addressed them in a point by point style below.

Review Comments:

- Fig 1b shows binding events at low Au-salt: there should also be (a few, but some) multiple steps?

Author Feedbacks:

We acknowledge the reviewer for this excellent comment. The reviewer is absolutely correct. Even at a low tetrachloroaurate(III) concentration ($<1 \mu\text{M}$), the observation of multi-step shaped event, which results from simultaneous binding from more than one tetrachloroaurate(III), is still possible due to the octameric symmetry of MspA. However, this probability has a clear concentration dependence, which is systematically demonstrated in **Fig. S9**.

From the trial that was demonstrated in **Fig. S9a, d and g**, within a 280 seconds continuous recording when the final concentration of tetrachloroaurate(III) in cis was $1 \mu\text{M}$, we didn't observe any event with multiple binding steps, as demonstrated in the histogram of **Fig. S9d**. However, the probability of observing the 2nd and the 3rd binding event would increase when a higher concentration of tetrachloroaurate(III) in cis was reached, as demonstrated in **Fig. S9b, e, h** and **Fig. S9c, f, i**.

Though negligible, the chance of observing multi-step event at a low concentration is still possible in principle. To further verify this, we have re-performed the measurement demonstrated in **Fig. 1b** and **Fig. S9a** in three independent trials. Within 10 min continuous recording, we did observe 1-4 two-step events from each trial, which accounts for 2%-3% of all events we have acquired. These results were listed in the table below:

Independent experiment	Total event counts	Multi-step event count	Single step event probability
1	127	4	97%
2	56	1	98%
3	86	3	97%

We acknowledge the reviewer for this wonderful comment. We have made corrections in the figure legend of **Fig 1** in the revised manuscript accordingly.

Review Comments:

The rhs shows a tenfold Au concentration with more (and multiple) steps. What happen at higher concentrations? Should there be a saturation? With 8 binding site in very close vicinity I expect some co-operativity.

Author Feedbacks:

We highly acknowledge reviewer 1 for this comment. Reviewer 1 is correct that at a higher concentration of tetrachloroaurate(III) in cis, it is imaginable that a higher probability of simultaneous tetrachloroaurate(III) binding within the same pore lumen may happen. However, from our measurements, the maximum number of simultaneous tetrachloroaurate(III) binding within the same pore is **three**. This is demonstrated in **Fig 1c**, which is also the same picture that reviewer 1 was mentioning about.

We have performed measurements with up to 40 μM tetrachloroaurate(III) in cis. Though the inter-event interval is decreased with a higher tetrachloroaurate(III) concentration in cis (**Fig. S9j**). We still observe simultaneous binding from only a maximum of **three** tetrachloroaurate(III). We believe that the space around the vicinity of the pore restriction may be too crowded to accommodate more than **three** tetrachloroaurate(III) simultaneously or a repulsion between tetrachloroaurate(III) may exist.

Though not published or included in this manuscript, our measurement with dichloroaurate(I) may also support our hypothesis. Similar to tetrachloroaurate(III), dichloroaurate(I) also binds to MspA-M, as observed from our ongoing research. However, we can observe simultaneous binding from up to **four** dichloroaurate(I) ions in the same pore (**RL Fig. 1**). This can be explained by the size difference of these two types of ions. Obviously, tetrachloroaurate(III), which is square planar in shape, requires a larger space when bound in the pore lumen than that required by a dichloroaurate(I) which is linear in shape.

RL Fig. 1. Electrophysiology recording with dichloroaurate(I) using MspA-M. Simultaneous binding from up to four dichloroaurate(I) can be monitored. I_4 indicates the event level when four dichloroaurate(I) were simultaneously bound to MspA-M.

Our result with dichloroaurate(I) indicates that simultaneous binding from 8 tetrachloroaurate(III) in the same pore requires a much bigger space in the pore lumen. Considering that the MspA nanopore is conical in shape, this may be achieved by placing the methionine residue slightly away from the pore restriction.

Eventually, we acknowledge reviewer 1 again for this comment and we have made corresponding explanations in the revised manuscript accordingly. However, we plan to report our measurement with dichloroaurate(I) in a separate further study so that the

topic of this manuscript is rather focused.

Review Comments:

How Fig 1e will look like at higher Au concentration?

Author Feedbacks:

We highly appreciate reviewer 1 for this comment. **Fig 1e** demonstrates that the inter-event interval has a concentration dependence. However, the event dwell time is independent of the analyte concentration. To avoid the complication resulted from possible multi-step binding events, the discussion in **Fig. 1e** was limited to measurements with a low tetrachloroaurate(III) concentration.

However, we did include results from measurements with higher tetrachloroaurate(III) concentrations, which has been demonstrated in **Fig. S9j** and may have been ignored by the reviewer. From **Fig S9j**, it demonstrates that the inter-event interval significantly decreases when a higher concentration of tetrachloroaurate(III) is placed in cis.

We acknowledge reviewer 1 again for the comment and we apologize for the confusion generated due to the fact that the MspA nanopore used in this study still has an octameric symmetry. However, this issue should be solved in our future construction of a hetero-octameric nanopore.

To avoid confusion for other readers, we have made corresponding corrections in the revised manuscript. The reader could also be guided to **Fig. S9** for other detailed results performed with a higher concentration of tetrachloroaurate(III).

Review Comments:

Could the binding site be a nucleation site for precipitation and Au particle growth?

Author Feedbacks:

We highly appreciate reviewer 1 for this wonderful question. According to reported literatures, cysteine or glutathione could bind strongly with gold nanoclusters (Journal of the American Chemical Society, 2007, 129(20): 6402-6404.; Colloids and Surfaces A: Physicochemical and Engineering Aspects, 2009, 338(1-3): 93-101.) or act as a reductant in the preparation of gold nanoparticles (Colloids and Surfaces A: Physicochemical and Engineering Aspects, 2008, 317(1-3): 229-233.). The reviewer is thus correct that an amino acid or a cluster of amino acids within the pore lumen could in principle act as a nucleation site for the precipitation of gold nanoclusters or nanoparticles.

Actually our original research motivation was to generate a layer or a cluster of gold deposition on the inner lumen of a biological nanopore. This concept is similar to that previously reported by Wanunu et al however with a solid state nanopore (Nano letters, 2013, 13(2): 423-429.). We unexpectedly discovered the phenomenon that WT α -HL would directly report binding of tetrachloroaurate(III). We then figured out the mechanism of this single molecule interaction and found it extremely interesting to be carried out in an engineered MspA instead. Then the rest of the story was clearly presented in the submitted manuscript.

However, according to the presented measurement configuration and the reagent used in this manuscript, we didn't observe the formation of gold nanoclusters or nanoparticles within the pore lumen. The reason is that the introduced methionine is not a strong reductant and there are only 8 methionine within the whole measurement space that can possibly act as reductant to generate gold nanoclusters.

To generate a binding/nucleation site within the pore lumen, it may be more straightforward to engineer MspA nanopore with a cysteine or a cluster of cysteines around its pore restriction, which may act as a reductant or may bind to a metal nanocluster (Colloids and Surfaces A: Physicochemical and Engineering Aspects, 2008, 317(1-3): 229-233.). The advantage of this approach is an atomic precision in pore engineering and new applications may be discovered afterwards. We acknowledge the reviewer for such a great advice and we are quite optimistic with it.

Review Comments:

Would pre-incubation lead to a more equilibrated system?

Author Feedbacks:

We appreciate reviewer 1 for the question. We believe that the reviewer is asking whether a pre-incubation of tetrachloroaurate(III) in the measurement chamber would generate more consistent single molecule binding kinetics during the nanopore measurement.

Experimentally, tetrachloroaurate(III) was placed in cis, followed with magnetic stirring and a short pre-incubation of ~1-2 minutes to reach a homogenous distribution before carrying out the measurement. The binding kinetics of tetrachloroaurate(III) binding doesn't show too much difference between that at the beginning of the measurement and that measured 2 hours later (**Fig. S10**). So we believe a short incubation such as 1-2 minute is enough to reach an equilibrated state.

We acknowledge the reviewer again for this comment. To address this issue, **Fig. S10** was added along with corresponding discussions in the revised manuscript.

Review Comments:

- Could the recordings with thiols be quantitative?

Author Feedbacks:

We acknowledge the reviewer for the constructive comment. The relative binding frequency of biothiol with respect to all $[\text{AuCl}_4]^-$ binding events is dependent on the final concentration of biothiols placed in trans, indicating that the recordings with biothiols could be quantitatively analysed. Corresponding discussions have been added in the revised manuscript and **Fig. S19** with more details.

Review Comments:

- Fig.2. the figures should be visible also in black and white, please use different symbols

Author Feedbacks:

We highly acknowledge the reviewer for this suggestion. We assume that the reviewer is mentioning **Fig. 2c, d and h**. In the revised manuscript, we have updated this figure with more distinguishable symbols. Now all items in **Fig. 2** should be visible in black and white. This suggestion is quite important in the re-production of the image when colour printing is not available. We sincerely apologize for being ignorant on this rather important issue and we sincerely acknowledge reviewer 1 for such a great advice.

Reviewer #2 (Remarks to the Author):

Review Comments:

This research successfully utilizes the alpha-hemolysin aHL and MspA protein channels to clearly demonstrate single molecule Au(III)-thioester and Au(III)-thiol coordination chemical reactions and combine these reactions to discriminate single molecule biothiol compounds L-cysteine, L-homocysteine and L-glutathione from. The first finding is that the anion tetrachloroaurate (AuCl_4^-) can selectively bind to the thioester of methionine constructed in the lumen of both protein pores. The reversible binding/release of a single AuCl_4^- to/from one methionine can be clearly read out from the nanopore conductance variation. The second finding is that MspA is superior compared with the aHL pore in term of AuCl_4^- binding efficiency, event conductance disperse, noise levels and blocking levels, due to the native conical shape lumen of the MspA pore that features a sharp sensing site at the trans opening. The third and the most important finding is that the Au(III) on the methionine can be further attacked by various biothiol compounds. When binding, the Au-thiol coordination with biothiol compounds are stronger than the Au-thioester bond with methionine, thus quickly remove the the Au-thiol compound from the methionine. In this reaction, the intermediate structure for methionine-Au(III)-thiol compounds can block the nanopore characteristically, thus allowing discriminate three biothiol compounds at the single molecule levels. This work is significant and has impact in at least two aspects: it demonstrates the capability of nanopore to dissect the mechanism for single molecule metal ion (e.g. gold)-involved chemical reaction with biological substances; and through manipulation of different Au-thiol ester and Au-thiol reactions to detect different biothiol compounds and potentially proteins.

The paper overall is suitable to publishing in this journal, but the authors still need to address several questions as follow

Author Feedbacks:

We acknowledge the reviewer for such positive overall feedbacks on our manuscript and we thank the reviewer for the recommendation of publishing. We also thank the reviewer for all questions raised, which are extremely inspiring to us. They are all addressed below in a point by point style.

Review Comments:

1. The authors firstly used the Au-thioester reaction to capture a tetrachloroaurate (III) (Au(III)) onto methionine on the nanopore, then used the Au-thiol reaction to capture biothiol compounds. The Au-thiol reaction can replace Au-thioester reaction, removing Au from methionine. To support this mechanistic hypothesis, it would be necessary to test the thioester compounds such as methionine. This free methionine in the solution should compete with the methionine on the nanopore to capture Au (III) on the binding site. This

reaction should be quite different from the Au-thiol reaction.

Author Feedbacks:

We highly acknowledge the reviewer for the extremely visionary question. Tetrachloroaurate(III) binds reversibly to the methionine residue placed around the pore restriction. It is reasonably expected that free methionine in the solution would compete with the methionine residue on the nanopore to bind to tetrachloroaurate(III).

Tentatively, we have pre-incubated 80 μM methionine and 4 μM tetrachloroaurate(III) both in cis and a measurement trace was acquired with a +100 mV continuously applied potential. We did observe single molecule phenomenon resulted from binding of free L-methionine to a Au(III) embedded nanopore (**RL Fig. 2**).

RL Fig. 2 L-methionine events acquired with Au(III) embedded MspA-M. **a**) A continuously acquired trace containing L-methionine events. State 1 indicates the open pore state. State 2 indicates the tetrachloroaurate(III) binding state. State 3 indicates the L-methionine binding state. **b**) Four types of L-methionine events. **c**) A suggested mechanism. **d**) Event height statistics for ΔI_{1-3} .

The reviewer is also correct in the prediction that this reaction is quite different from the case with biothiols.

First, binding events from free L-methionine is more frequently observed when tetrachloroaurate(III) and L-methionine were both placed in the cis chamber. However, by placing tetrachloroaurate(III) in cis and L-methionine in trans with a 4 μM and 40 μM concentration separately, these events don't appear as efficiently as that observed from biothiols. These results indicate that the free L-methionine doesn't react with the bound Au(III) atom as rapidly as the biothiols.

Second, binding events of free L-methionine results in flat, long residing plateaus (**RL**

Fig. 2b), however binding events of biothiols results in more fluctuating, shorter residing signals (**Fig. 3e**).

Third, binding events of free L-methionine appear as four types of events (**RL Fig. 2b**). However, binding of biothiols normally appear after the appearance of the tetrachloroaurate(III) binding event, indicating that all biothiols tested in this paper bind with Au(III) much more stronger than the methionine residue on the pore.

To summarize, as commented by the reviewer, we did observe competitive binding from free L-methionine to a bound Au(III) atom against the methione on the pore. However, the competition from biothiol could be much more efficiently observed, which suggests direct biothiol sensing as the main application of this technique. The reviewer is also correct that binding events from L-methionine appear quite differently from biothiols.

However, we didn't include results of L-methionine sensing in this manuscript so that the topic of this paper is more focused. We acknowledge the reviewer again for such a constructive suggestion, which definitely worth a systematic study in our follow-up research.

Review Comments:

2. Au(III) binding to methionine on the nanopore surface is a one-step reaction. This can be reflected by the one-step nanopore current blockade. In contrast, the binding of biothiol compound to the bound Au(III) is a multi-step binding, as reflected by the dramatic multi-level current variation at the end of each event. These are intermediate states. Can the authors discuss this difference between the two types of Au-S reactions and their nanopore signatures.

Author Feedbacks:

We highly acknowledge the reviewer for this fantastic and visionary question. The reviewer is absolutely correct that binding of biothiol and the stimulated release of Au(III) from MspA-M contains rich information of intermediate states, different from a rather simple Au(III)-thioether interaction.

To compare these two states, Au(III)-thioether interaction results in a simple one step binding and release, appearing as a flat, low noise event step. On the other hand, Au(III)-thiol interaction results in significant current fluctuations. This could be explained by the fact that the thiol group of a biothiol molecule is a much stronger reductant than the thioether group of a methionine. According to reported literatures and existing knowledges in bioinorganic chemistry, the interaction between a tetrachloroaurate(III) and a methionine is mainly reversible, non-redox coordination interactions. On the other hand, thiol tetrachloroaurate(III) interaction involves redox reactions, which should contribute to the noise observed.

We consider the comment from the reviewer a great suggestion to improve the way that we present our manuscript. Thus, we have included more discussions on the characteristics of the event shape in the revised manuscript. Besides, it is inspiring for us that future research may be carried out to fully understand the chemical nature of the observed intermediate states.

Reviewers' Comments:

Reviewer #1:

Remarks to the Author:

The authors improved the ms. sufficiently and can be published as it is.

Reviewer #2:

Remarks to the Author:

I have asked two main questions about the Au-thiol reaction mechanism revealed by the MspA protein pore. The authors have answered these questions by supplying detailed experimental supports.

Overall, I agree to publish this single molecule chemistry research report.

A minor suggestion is that the new result on the native methionine competitive binding for question 1 is also suitable to be placed in Supplementary Materials. But it subjects to the authors.

Response Letter

Reviewer #1 (Remarks to the Author):

Review Comments:

The authors improved the ms. sufficiently and can be published as it is.

Author Feedbacks:

We acknowledge reviewer 1 for all previous comments and your suggestion for publishing.

Reviewer #2 (Remarks to the Author):

Review Comments:

I have asked two main questions about the Au-thiol reaction mechanism revealed by the MspA protein pore. The authors have answered these questions by supplying detailed experimental supports.

Overall, I agree to publish this single molecule chemistry research report.

A minor suggestion is that the new result on the native methionine competitive binding for question 1 is also suitable to be placed in Supplementary Materials. But it subjects to the authors.

Author Feedbacks:

We acknowledge reviewer 2 for all previous comments and your support for publishing.

The suggestion to place materials of native methionine competitive binding in the revised manuscript is fantastic as it may help to demonstrate another sensing application of Au(III) embedment in MspA. Though we have tentatively tried to place these materials in the revised SI, due to the extremely crowded space in the manuscript to mention and discuss the native methionine competitive binding phenomenon, it is rather difficult for us to squeeze these materials in anymore. On the other side, without placing these materials in the manuscript, the main topic of the whole manuscript is more focused.

Again, we acknowledge reviewer 2 for such precious comment and for allowing us to decide whether these materials may be placed in the manuscript or not. We noticed that when publishing in Nature Communications, all review comments are published as a separate, standalone supplementary file. Thus, we decide not to place these materials in the revised manuscript. However, results of native methionine competitive binding along with its discussions are fully disclosed to interested readers anyway.